# Genetic stock structure of the silky shark *Carcharhinus falciformis* in the Indo-Pacific Ocean

Chia-Yun Joanne Li[1], Wen-Pei Tsai[2], R. R. M. K. P. Ranatunga[3], Munandar Samidon[4], Shang Yin Vanson Liu[1]*

**1** Department of Marine Biotechnology and Resources, National Sun Yat-sen University, Kaohsiung, Taiwan, **2** Department of Fisheries Production and Management, National Kaohsiung University of Science and Technology, Kaohsiung, Taiwan, **3** Centre for Marine Science and Technology, Department of Zoology, University of Sri Jayewardenepura, Nugegoda, Sri Lanka, **4** Department of Marine Science, Teuku Umar University, Aceh Barat, Indonesia

* oceandiver6426@gmail.com

**Data Availability Statement:** All relevant data are within the paper and its Supporting Information files.

## Abstract

The silky shark, *Carcharhinus falciformis*, is a cosmopolitan species commonly caught as a bycatch for longline fisheries. However, the genetic stock structure for the Indo-Pacific Ocean is not well-defined yet. Here, we used eight microsatellite loci to examine the genetic stock structure and effective population size of 307 silky sharks across 5 Indo-Pacific sampling locations. A major genetic break was found between Aceh and the remaining locations ($F_{ST}$ = 0.0505–0.0828, p = 0.001). The Indian Ocean displayed a slightly lower effective population estimate (Ne) compared to the Pacific Ocean, potentially due to the higher fishing pressure in the Indian Ocean region. The lowest Ne was found in the Aceh population (Ne = 2.3), suggesting it might be a small and endemic population. These findings offer valuable information for the conservation and management of the silky shark. We suggest that the population around Aceh waters constitutes a distinct stock and should be managed independently. Further investigations into migratory and movement patterns are needed to define the boundaries of different stocks, ensuring effective management the silky shark across the Indo-Pacific region.

## Introduction

Elasmobranchs play an important role in the marine ecosystem, serving as apex predators to stabilize the ecosystem and biodiversity. Due to their K-selected life history traits, including slow growth, late maturity, low fecundity and long lifespan, elasmobranchs are more susceptible to overfishing and have lower recovery potential compared to r-selected bony fish [1, 2]. High global demand for shark products, such as fins, skins, gill plates, liver oil, and jaws, has led to over 391 elasmobranch species being categorized as threatened by the International Union for Conservation of Nature (IUCN) [3]. Numerous shark species have also been listed in Appendix I and Appendix II of the Convention on International Trade in Endangered

**Funding:** This study was funded by Ministry of Science and Technology, Taiwan (MOST) under No. MOST110-2313-B-992-001, No. MOST107-2911-I-110-301. The funders had no role in study design, data collection and analysis, decision to publish, or preparation of the manuscript

**Competing interests:** The authors have declared that no competing interests exist.

Species of Wild Fauna and Flora (CITES), restricting their international trade in response to declining populations over the past few decades [4, 5]. This situation underscores the urgent need for comprehensive management strategies to ensure the future sustainable use of wild population.

To effectively manage widely dispersed marine resources, understanding the management or stock units across their distribution is crucial. The genetic stock and connectivity of highly mobile marine species, like sharks and rays, can be influenced by a range of factors, including habitat preference, movement patterns, reproductive strategies, and environmental factors [6]. Oceanic sharks, like the blue shark *Prionace glauca* and the basking shark *Cetorhinus maximus*, renowned for their impressive migratory capabilities, demonstrate little to no differentiation in population structure across ocean basins [7–9]. In contrast, coastal-dwelling sharks such as the grey reef shark *Carcharhinus amblyrhynchos* and the gummy shark *Mustelus antarcticus* tend to exhibit more pronounced population structures within relatively localized regions [10, 11]. Behaviors such as sex-biased dispersal [12, 13], site fidelity [14], and female philopatry [15] can also impact the degree of population structure and contribute to sex-biased gene flow between populations. This information is crucial for choosing management strategies that align with the scales and types of fishing pressure across different regions.

The silky shark, *Carcharhinus falciformis*, is widely distributed in tropical and subtropical oceans [16, 17]. Depending on their life stage, it is commonly found in pelagic and coastal regions. Adults and older juveniles are widely found in deep waters off continental and insular shelves, while neonates and early juveniles inhabit deeper areas of the continental shelf [16, 18]. The silky shark is also known for its highly migratory nature, with documented travel distances of up to 1,339 km in the western North Atlantic and 2,200 km in the eastern tropical Pacific Ocean region [19, 20]. While there is no solid evidence to support sex segregation and female philopatry behavior in the silky shark [21–23]. It is among the most exploited shark species and is commonly caught as bycatch in tuna longline and purse seine fishery due to its overlapping presence with tuna school [16, 24, 25]. Additionally, there is a high demand for its fins, which are abundantly traded in markets in Guangzhou, Hong Kong, and Taiwan [26–29].

To date, the silky shark has been listed as a 'Vulnerable' species on IUCN Red List [30] and as an Appendix II species under CITES to regulate its international trade. Stock assessments indicated a significant decline, especially in the Atlantic and Western Indo-Pacific oceans [31, 32]. Inter-governmental fishery organizations, such as the International Commission for the Conservation of Atlantic Tuna (ICCAT) and the Western and Central Pacific Fisheries Commission (WCPFC) have prohibited retaining the silky shark [33, 34]. In the Eastern Pacific, retention bans are enforced only for purse seine fisheries by the Inter-American Tropical Tuna Commission (IATTC). There is no regulation on the silky shark catch in the Indian Ocean due to insufficient information about the species. This inconsistency in regulations across its range underscores the urgent need to understand its stock structure both among and within oceans.

Until now, studies have primarily used mitochondrial DNA (mtDNA) genetic markers to investigate the genetic population of the silky sharks at different scales. For instance, Galván-Tirado et al. (2013) suggested, based on mtDNA genetic variation, that populations in the East and West Pacific should be managed as two stocks [35]. Clarke et al. (2015) identified at least five matrilineal populations on a global scale [36]. However, since mtDNA is maternally inherited, there is potential for bias when inferring genetic connectivity, especially if the species under study exhibits sex-biased dispersal. In contrast, nuclear markers, which have been effectively used to discern the genetic structure of sharks (i.e., the blackmouth catshark *Galeus melastomus*) [37], offer a valuable tool for stock identification studies. Recently, Kraft et al. (2020) utilized single nucleotide polymorphisms (SNPs) data to detect the genetic structure of the silky shark in both the Atlantic and Indian Oceans, successfully identifying a fine-scale genetic

structure between populations [38]. Given these findings, there is a compelling case for employing nuclear markers more extensively to achieve a clearer understanding of the contemporary genetic stocks of the silky shark. Additionally, genetic data can serve as an effective tool for estimating the effective population size (Ne) - a important parameter in wildlife monitoring and management [39]. Ne represents the Wright-Fisher idealized population size, which is determined by the changes in gene frequency variance or the rate of inbreeding [40].

In the present study, we utilized eight microsatellite markers to identify population differences among five putative silky shark populations spanning the Western Pacific and the Western Indian Ocean. Additionally, we aimed to uncover their effective population sizes. Our findings offer enhanced insight into the stock structure of the stock structure of the silky shark over a broad geographic range, especially in the Indian Ocean where fisheries remain active. This information is invaluable for future management and conservation initiatives.

## Materials and methods

### Ethics statement

Samples used in this study were from fish landings; the specimens were no longer alive when we collected tissue samples for population genetic analysis. Thus, no permission is needed from Institutional Animal Care and Use Committee (IACUC).

### Sampling

A total of 307 tissue samples were collected from 5 locations, including the Southwestern Pacific Ocean (PO, N = 116), Indonesia Aceh fish market (Aceh, N = 41), Sri Lanka Negombo and Colombo fish market (SLW, N = 68), Mirissa and Kudawella fishing ports (SLS, N = 40), and Western Indian Ocean (OB, N = 42) (Fig 1 generated by ggOceanMap in R with Natural Earth Data) from September 2018 to January 2021. The sampling map was generated using ggOceanMaps package and ggplot2 package [41, 42]. Dr. Ranatunga and Dr. Samidon collected tissue samples from Sri Lanka and Indonesia, respectively. In the Western Indian Ocean, samples of the silky shark were opportunistically collected by fishery observers aboard Taiwanese large-scale tuna longline vessel during the fishing operations. Species identifications relied on morphological features, including a small, low second dorsal fin, a low ridge between the dorsal fins, and elongated free rear tips of second dorsal and anal fins. Tissue samples (1 cm$^3$) from fins and muscles were immediately preserved in the field in 95% ethanol for further DNA analysis.

### DNA extraction and genotyping

DNA was extracted from muscle or fin tissue using a Genomic DNA extraction kit according to the manufacturer's recommended protocol (Genomics BioSci. And Tech. Co., Taiwan). A total of 8 microsatellite markers were selected from O'Bryhim et al. (2015) and used for genotyping [43]. Primers were labeled with either FAM or TAMRA fluorescent dye (Genomics, Taiwan). Microsatellites were amplified in 25uL reactions with the following profile: 95˚C for 4 min, followed by 36 cycles of denaturation at 94˚C for 30 s, annealing at 55–60˚C for 30 s, and extension at 72˚C for 30 s, and a final extension step at 4˚C for 1 min. Each reaction contained 50ng DNA, 12.5μL TaqDNA polymerase 2X master Mix RED (0.4 mM each dNTPs, 15 mM MgCl, 0.2 unit of Ampliqon DNA polymerase) (Ampliqon Denmark) and each of the primers (200nM). Genotyping was conducted using an ABI sequencer and alleles sizes were analyzed with GeneMapper Software V.4.1 (Applied Biosystem, USA).

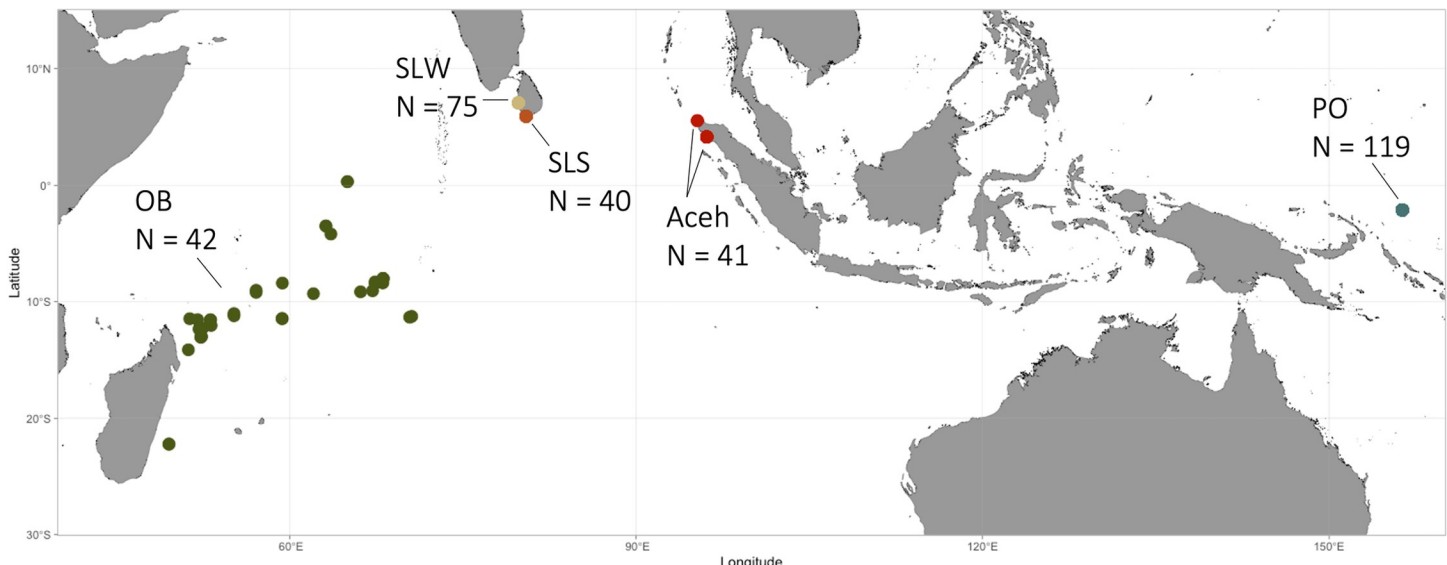

**Fig 1. Map of five sampling sites in this study, the Pacific Ocean area is designated as "PO", Indonesia Aceh fish markets are designated as "Aceh", Sri Lanka Negombo and Colombo fish markets are indicated by SLW, Mirissa and Kudawella fishing ports are designated as SLS, and Western Indian Ocean area is designated as OB.**

### Genetic diversity analyses

Mean private allele ($A_P$), observed heterozygosity ($H_o$), expected heterozygosity ($H_E$), and inbreeding coefficient ($F_{IS}$) were computed in GenAlEx 6.5 [44]. Mean allelic richness ($A_R$) was computed using a rarefaction method in hierfstat package [45] in R, as this method accounts for differences in sample size. Significant deviations from Hardy-Weinberg equilibrium (HWE) at each locus and linkage disequilibrium (LD) between pairs of loci were tested by running 10,000 Markov chain iterations in the Genepop package [46] in R and significant p-value was adjusted using a sequential Bonferroni correction.

### Population structure analyses

Pairwise $F_{ST}$ value with Fisher's exact test was estimated using MSA 4.05 [47]. The Bayesian assignment test was performed in STRUCTURE v2.3 [48] under an admixture model with sampling locations (K = 5) as priors to investigate population structure. The parameters were set as follows: a burn-in period of $10^5$ iterations, $10^6$ recorded iterations for K = 1 to K = 5, and 10 replicates of each K value. The most likely number of clusters was computed using Evanno's ΔK approach with Structure Harvester online [49]. The repetition for the best K was aligned, merged, and visualized using pophelper package [50] in R.

Population structure was also assessed with Discriminant Analysis of Principal Components (DAPC) for five putative populations in the Adgenet v1.3.4 package [51] in R. Firstly, using function *find.clusters* ran successive K-means, following identified the optimal cluster number based on the Bayesian Information Criterion (BIC) of the corresponding model, and finally, using *xvalDapc* to determine optimal PCs numbers.

Analysis of Molecular Variance (AMOVA) was computed in pegas package [52] in R, which was used to detect the population differentiation based on three grouping hypotheses: (i) geographic group, (ii) sampling location, and (iii) structure result.

## Barrier analysis

The barrier analysis was applied to detect the genetic boundary between the population based on Monmonier's (1973) algorithm [53] function in Adgenet v1.3.4 package [51] in R which detects boundaries among geographic populations with both geographic and genetic distances as a valuated graph. This is achieved by finding the path exhibiting the largest distances between connected populations. The genetic distances between neighboring groups were calculated based on microsatellite data and coordinates. After finding the most significant genetic distance between two sampling locations, the algorithm extends the boundary line to the following largest genetic distance of the neighboring location. Finally, the *optimizer.monmonier* function was used to detect the largest sum of distances to interpret genetic distances among populations. Results of barrier analysis were illustrated using ggOceanMap and ggplots2 package in R [41, 42].

## Effective population size (Ne)

Contemporary effective population size was computed using the single-sample molecular method - Linkage Disequilibrium method as implemented in NeEstimator v2.0 [54], which was estimated from the amount of pairwise linkage disequilibrium between microsatellite loci. According to Waples and Do (2010), harmonic means of Ne were used to avoid skewed negative Ne [55]. The lowest allele frequency ($P_{crit}$) value is the criterion for excluding rare alleles from the analysis.

## Results

### Genetic diversity

The genetic diversity indices given in Table 1 are based on 307 multilocus genotypes (8 microsatellite loci). The genotype data for the 307 silky shark samples were provided in S1 Table. The mean observed heterozygosity ($H_O$) for all loci in each location ranged from 0.51 (PO) to 0.71 (Aceh). Expected heterozygosity ($H_E$) varied from 0.73 (PO) to 0.81 (SLW). Mean allele richness ($A_R$) ranged from 6.18 (Aceh) to 10.68 (SLW). The average private allele varied from 0.38 (OB) to 2.63 (SLW). The inbreeding coefficient ($F_{IS}$) was between 0.17 (SLW) and 0.28 (SLS) and was statistically significant in most sampling locations ($p < 0.05$) (Table 1). The HWE test results indicated that most populations, particularly at loci Cafa16 and Cafa44, deviated from HWE. Significant LD mainly occurred in Aceh across all loci (S2 and S3 Tables). To assess the potential influence of Cafa16 and Cafa44, we excluded these two loci and recalculated pairwise $F_{ST}$, assignment test and Ne estimation. The results showed similar patterns with those based on 8 loci (S4 and S5 Tables and S1 Fig).

**Table 1. Summary of genetic diversity statistics derived from 5 sampling sites from 8 microsatellite loci.**

| | Sampling site | N | $H_O$ | $H_E$ | $A_R$ | $A_P$ | $F_{IS}$ |
|---|---|---|---|---|---|---|---|
| **Pacific Ocean** | **PO** | 116 | 0.51 | 0.73 | 9.24 | 2.38 | 0.27 |
| **Indian Ocean** | **Aceh** | 41 | 0.71 | 0.73 | 6.18 | 1.00 | 0.02 |
| | **SLS** | 40 | 0.56 | 0.78 | 8.81 | 0.63 | 0.28 |
| | **SLW** | 68 | 0.66 | 0.81 | 10.68 | 2.63 | 0.17 |
| | **OB** | 42 | 0.61 | 0.77 | 8.98 | 0.38 | 0.19 |

N = sample size; $H_O$ = observed heterozygosity; $H_E$ = expected heterozygosity; $A_R$ = allelic richness; $A_P$ = private allele; $F_{IS}$ = inbreeding coefficient.

## Major genetic structure

Overall, the pairwise $F_{ST}$ value among the five sampling locations ranged from 0.0045 (OB vs. SLW) to 0.083 (PO vs. Aceh) (Table 2). The results of the assignment test showed that K = 2 has the highest $\Delta$ K value, as determined by STRUCTURE harvester results. The $F_{ST}$ value between the Pacific Ocean and Aceh was the highest and significant ($F_{ST}$ = 0.083, p < 0.001). In the DAPC scatterplot, Aceh also showed clear separation from the rest of the sampling locations, which indicated that Aceh was genetically isolated from the rest of the groups (Fig 2).

Regarding the STRUCTURE bar plot, the genetic composition of the Aceh population was clearly different from the rest of the populations (Fig 3). This pattern was congruent with the results of $F_{ST}$ test and DAPC plot.

Finally, AMOVA revealed that 2.61%, 3.40%, and 5.53% of the genetic variation occurred between populations (Table 3) in the geographic grouping, sampling location, and STRUCTURE clustering hypothesis, respectively. The STRUCTURE clustering hypothesis explained the highest genetic variation between population levels among the three hypotheses.

## Minor genetic structure

Pairwise $F_{ST}$ comparisons between the Pacific Ocean and the rest of the Indian Ocean populations (excluding Aceh) were significant, indicating restricted gene flow between the Pacific Ocean and the Indian Ocean ($F_{ST}$ = 0.212–0.336, p = 0.001) (Table 2). This genetic structure was also observed when excluding the Aceh population under K = 2 scenario. Among the ten total pairwise $F_{ST}$ comparisons, two showed no significance (OB vs. SLS and OB vs. SLW) with relatively low $F_{ST}$ values. Notably, the pairwise $F_{ST}$ value between SLW and SLS was low but significant ($F_{ST}$ = 0.0098, p <0.001) (Table 2). The DAPC scatterplot depicted overlapping ellipses for SLS, SLW, and OB, while PO was slightly separated.

Two significant genetic barriers were found using monmonier's algorithm in the North Indian Ocean and the South Indian Ocean, respectively. The second barrier separated Sri Lanka and the Observer sample. Finally, slight boundaries were observed within the Observer group (Fig 4).

## Effective population size (Ne)

To analyze contemporary Ne, we separated individuals into three groups based on the genetic partitions we found through several population genetic analyses based on eight microsatellite loci: (i) the Pacific Ocean (PO); (ii) Indonesia (Aceh), and (iii) the Indian Ocean (SLS, SLW, and OB). To exclude rare alleles frequencies from analysis, only $P_{crit}$ = 0.02 was considered. Due to some missing data, the harmonic mean sample size from the Pacific Ocean, Aceh, and the Indian Ocean were 115.5, 39.9, and 148.1, respectively (Table 4). While $P_{crit}$ = 0.02, the Ne was 617.7 for PO (95% $CI_{parametric}$: 281.6-Infinite), 2.3 for Aceh (95% $CI_{parametric}$: 2.0–2.6), and 398.8 for the Indian Ocean (95% $CI_{parametric}$: 244–932.0).

**Table 2. Pairwise $F_{ST}$ value among the 5 sampling sites (below diagonal) and corresponding p values (above diagonal).**

|      | PO     | Aceh   | SLS    | SLW    | OB     |
|------|--------|--------|--------|--------|--------|
| PO   | -      | 0.001  | 0.001  | 0.001  | 0.001  |
| Aceh | 0.0828 | -      | 0.001  | 0.001  | 0.001  |
| SLS  | 0.0212 | 0.0582 | -      | 0.001  | n.s.   |
| SLW  | 0.0336 | 0.0505 | 0.0098 | -      | n.s.   |
| OB   | 0.0234 | 0.0569 | 0.0100 | 0.0045 | -      |

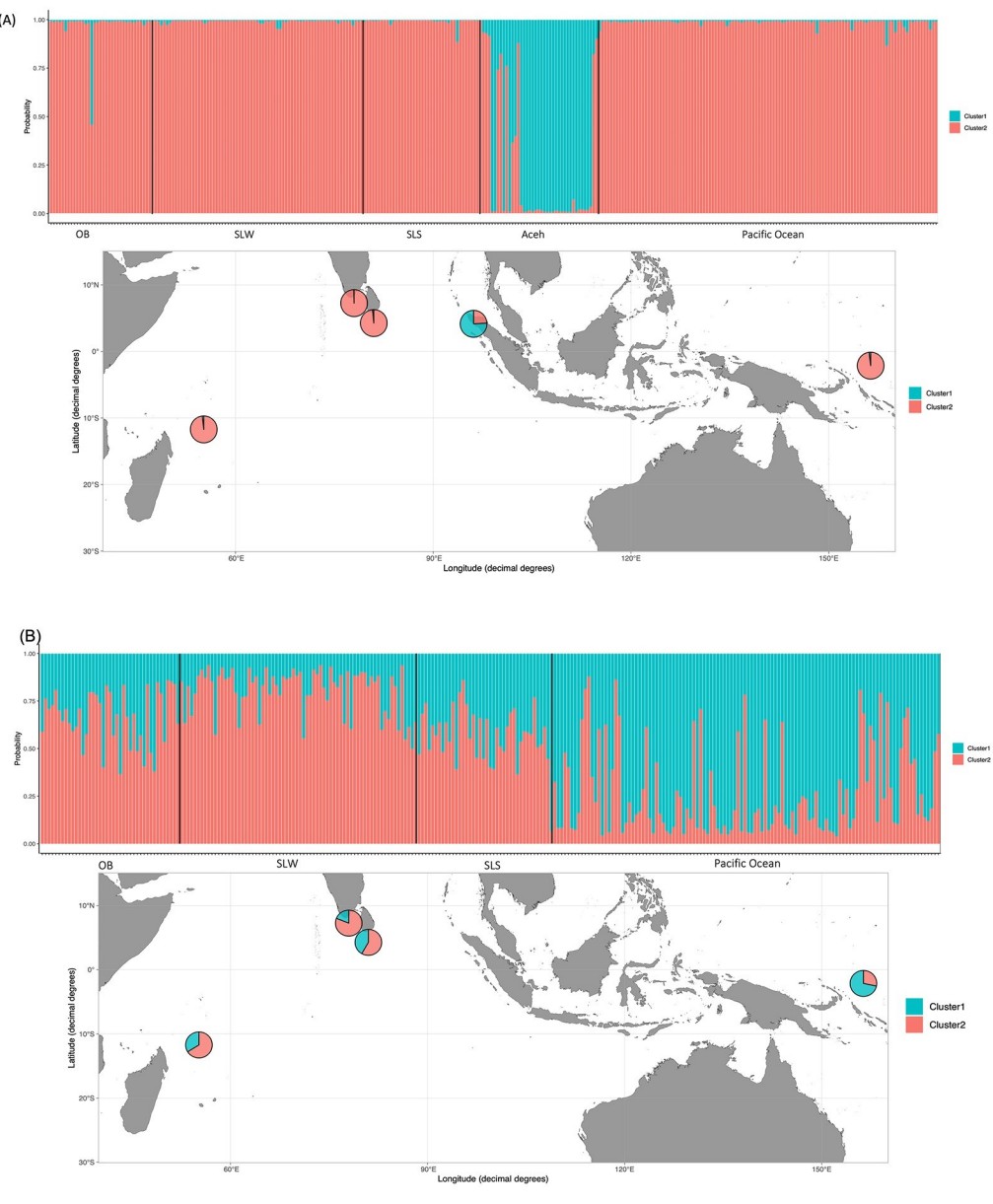

**Fig 2. Scatterplot of DAPC genotyped at 8 microsatellite loci.**

## Discussion

This is the first study using microsatellite loci to identify the genetic stock structure of the silky shark across the Indo-Pacific region. The results reveal (i) a strong population structure exists between Aceh and the rest of the population and (ii) a minor genetic structure was found between Indian and Pacific populations and within Sri Lanka samples. The results of this study provide crucial information for the future management of the silky shark, particularly in regions where commercial fishing is still permitted, such as the Indian Ocean.

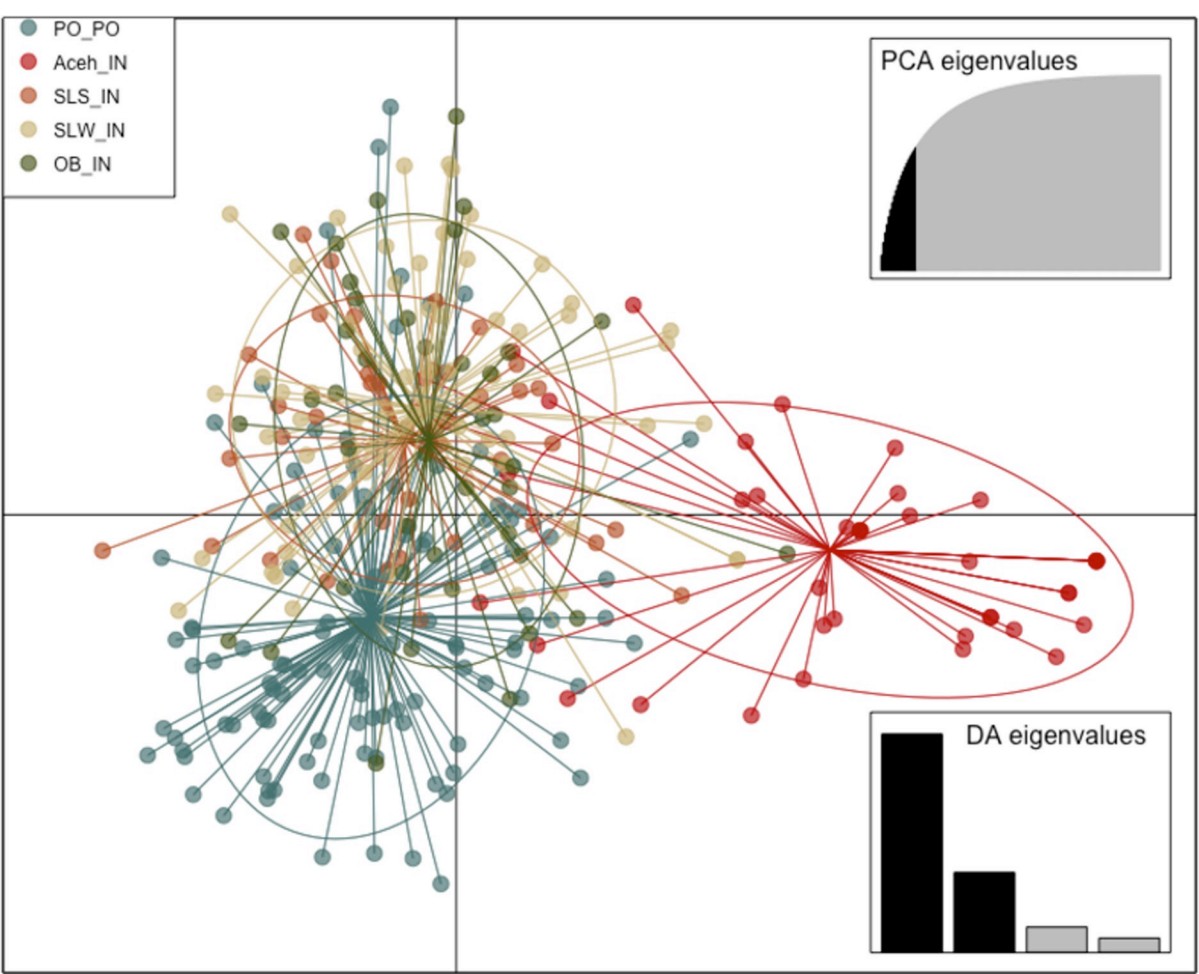

**Fig 3.** Structure plots showing the most likely number of clusters: (A) All sample sites included K = 2; (B) 4 sample sites excluding Aceh, K = 2.

## Population genetic structure

Previous studies on the silky shark have typically utilized mtDNA as a genetic marker [35, 36, 56]. Their finding have identified at least six distinct genetic lineages on a global scale: (i) Eastern Pacific Ocean; (ii) Western Pacific Ocean; (iii) Indian Ocean; (iv) Red Sea; (v)

**Table 3. Analysis of molecular variance (AMOVA) partitioning genetic structure based on three grouping hypotheses.**

| Hypothesis tested | Source of Variance | % of variation | Phi ST | P-value |
|---|---|---|---|---|
| Geographic Group (PO v.s. Aceh, SLS, SLW, OB) | Between pop | 2.607 | 0.265 | 0.001 |
| | Between subpop Within pop | 23.923 | 0.246 | 0.001 |
| | Within samples | 73.470 | 0.026 | 0.001 |
| Sampling Location (PO v.s. Aceh v.s. SLS v.s. SLW v.s. OB) | Between pop | 3.398 | 0.261 | 0.001 |
| | Between samples Within pop | 22.721 | 0.235 | 0.001 |
| | Within samples | 73.881 | 0.034 | 0.001 |
| Structure cluster result (Aceh v.s. PO, SLS, SLW, OB) | Between pop | 5.530 | 0.287 | 0.001 |
| | Between samples within pop | 23.137 | 0.245 | 0.001 |
| | Within samples | 71.333 | 0.055 | 0.001 |

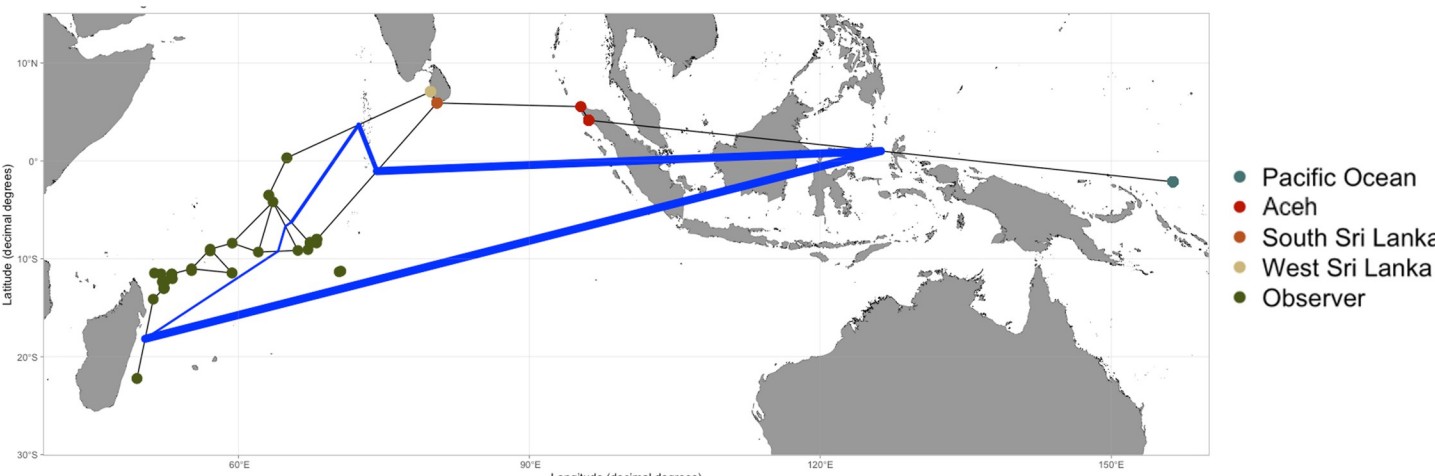

**Fig 4. Genetic barriers (blue lines) were predicted using Monmonier's algorithm with the Gabriel graph.** The black lines represent the gene-flow paths allowed by the models. The thickness of blue lines represented the levels of the effect of barriers on gene flow.

Northwestern Atlantic Ocean; and (vi) Southwestern Atlantic Ocean [36, 56]. However, recent research by Kraft et al. (2020) introduced the pool-seq method to investigate populations in the Atlantic Ocean and Red Sea. This approach revealed previously undetected population structures among inter-Atlantic regions, specifically in the Gulf of Mexico, Western Atlantic, and along the Brazilian coast, which had not been identified using mtDNA analysis as reported by Clarke et al. (2015). Consequently, the use of highly variable genetic markers like SNPs and microsatellites may offer higher resolution in revealing fine-scale genetic structures. Furthermore, the maternally inherited nature of mtDNA can limit the detection of gene flow, especially when there is sex-biased dispersal [15]. Therefore, in the present study, we employed polymorphic microsatellite loci to elucidate the contemporary genetic stock structure in the Indo-Pacific region. Our results revealed the presence of at least two genetic stocks across the sampling locations. Notably, the Aceh population stood out as a unique genetic stock, as indicated by the results of STRUCTURE and DAPC analyses (Figs 2 and 3A). Additionally, the significant pairwise $F_{ST}$ values between Aceh and the other populations suggest that the silky shark population in this region is isolated from those in the Indian Ocean and the Western Pacific Ocean. This finding is further supported by the results of monmonier's algorithm, which highlights a genetic barrier between Ache and the populations of the Pacific Ocean and Western Indian Ocean.

Banda Aceh has one of the largest shark and ray fisheries markets in Indonesia [57]. This study collected silky shark samples from two landing sites: Banda Aceh and Meulaboh. The dominant fisheries in these areas include coastal and pelagic fisheries operating in the Indian

**Table 4. Effective population size (Ne) for three groups: PO = Pacific Ocean; Aceh; IN = Indian Ocean (SLS, SLW, OB) with $P_{crit} = 0.02$.**

|  | N | Estimated Ne | 95% CIs (parametric) |
|---|---|---|---|
| PO | 115.5 | 617.7 | 281.6-Infinite |
| Aceh | 38.5 | 2.3 | 2.0–2.6 |
| IN | 147 | 398.8 | 244.0–932.0 |

N = Harmonic Mean sample size.

Ocean, Malacca Strait, and Andaman Sea [57, 58]. Sharks in these regions are caught both as bycatch and target species in tuna longline and handline fisheries [59]. In our study, biological data such as sex and total length were not obtained during sampling (sampling from processed fish fillets). However, Dharmadi et al. (2017) discovered that the body lengths of the silky shark caught from tuna longline/handline operations and landed in Banda Aceh mainly ranged from 71 to 130 cm, with an approximately 1:1 sex ratio [59]. Considering the smaller body size of silky sharks caught in Aceh coastal waters and the prevalent use of traditional handline fishing practices in this region, we suspect that this region may serve as a nursery ground of an endemic stock of the silky shark.

Besides biological features, environmental conditions, ocean currents, distance, and physical barriers have also been considered as factors influencing the population structure of marine organisms [60–62]. The most commonly mentioned mechanism driving population structure is 'isolation by distance' (IBD), which demonstrates that genetic distance between populations positively correlates with their geographic distance [63]. However, in the present study, we did not perform the IBD test since the number of pairwise comparisons was too small to provide sufficient statistical support for the result. Additionally, we identified a significant genetic barrier in the middle of the Indo-Pacific region, suggesting that IBD may not be the main mechanism driving the observed genetic pattern. The current regime could also play a crucial role in influencing the genetic structure of marine organisms in the Coral Triangle [64]. The Indonesian throughflow current acts as a genetic barrier for the zebra shark *Stegostoma fasciatum* within the Indo-Pacific region [65]. However, there is no unique hydrodynamic regime (e.g., eddy or strong coastal current) observed along the coastal area of Aceh. Therefore, the current is unlikely to be the sole reason for the observed silky shark population structure in the present study. Among the potential mechanisms considered during this study, isolation followed by secondary contact appears to be the most influential mechanism of genetic partitioning in the silky shark. The effect of the low sea-level stands during glacial periods likely restricted gene flow between Pacific and Indian Ocean populations, resulting in genetic partitioning [10, 66, 67]. After the glacial retreat, the barriers to gene flow disappeared, allowing two separated populations to potentially come into secondary contact. This scenario often lead to genetic admixture at secondary contact zones [68], a phenomenon observed in many marine fishes as well as in our finding in Aceh (Fig 3B) [69, 70]. In bony fish such as frigate tuna *Auxis thazard* and skipjack tuna *Katsuwonus pelamis*, divergence between Aceh and Eastern Indonesia has also been observed [71]. The exposure of the Sunda Shelf during Pleistocene low sea stands was suggested as the cause of population divergence. In addition, since Aceh is located in the westernmost boundaries of the Indo-Pacific barrier, intraspecific isolation among demes is expected in this region [71]. Secondary contacts between populations often result in high local genetic diversity [72], as we observed in the heterozygosity in Aceh (Ho = 0.71). Therefore, we propose that the population structure of the silky shark within the Indo-Pacific Ocean is likely the result of isolation followed by secondary contact.

Surprisingly, the results of the pairwise $F_{ST}$ test revealed a moderate and significant difference between west Sri Lanka (Negombo and Peliyagoda) and south Sri Lanka (Mirissa and Kudawella). The difference may be explained by the variations in fishing strategies between west and south regions of Sri Lanka. In west Sri Lanka, sharks were primarily caught in coastal areas, whereas in south Sri Lanka, a greater variety of pelagic shark species are found [73, 74]. It is possible that the silky shark landing in south Sri Lanka originate from the southern Indian Ocean. Moreover, both SLS and SLW showed no significant difference from the western Indian Ocean (OB) sample. This observation might indicate a high level of genetic connectivity between the west and east Indian Ocean or suggest that Sri Lankan fishing fleets operate in both the western and eastern Indian Ocean. These major and minor genetic structures that we

observed highlight the effectiveness of using microsatellite loci in revealing regional genetic structures between ocean basins and finer genetic structures within basins.

## Effective population size

Effective population size is an important parameter for conservation biology. However, assessing the population size of pelagic species on-site, using direct approaches such as mark-recapture and satellite tagging, is often challenging [75, 76]. Consequently, the molecular approach has become a popular tool for estimating effective population size, offering greater convenience when conducting Ne estimates indirectly [77]. Ne is widely used in fishery assessment, wildlife monitoring, and management as a relative index of fish abundance and stock status. In our result, the Ne of the Aceh population (Ne = 2.3) is much lower than the Pacific Ocean (Ne = 617.7) and the Indian Ocean (Ne = 398.8) populations. Serval factors may have influenced the Ne estimated in the study, including the smaller sample size used for Aceh (N = 41), which could potentially lead to an underestimation of Ne [78], and the potential bias introduced by pooling samples from different geographic locations (Indian Ocean population). Therefore, the Ne estimated in our study may not precisely represent the true Ne; rather, it is a relative Ne value estimated based on the genetic variations within each population. This suggests that the silky shark inhabiting Aceh waters may constitute an isolated and endemic population. This is further supported by the deviation from LD for all loci in the Aceh population, which could be influenced by several factors, including selection, genetic drift, inbreeding, demographic history and population subdivision [79]. However, this situation is unlikely to be caused by natural selection and inbreeding in our case since the inbreeding coefficient of Aceh was the lowest ($F_{IS}$ = 0.02), suggesting that it could be a population with random mating. Theoretically, individuals in small and isolated populations tend to inbreed [80]. Nevertheless, in some sharks (e.g., the sicklefin lemon shark *Negaprion acutidens* and the basking shark), specific mating behaviors have evolved to avoid inbreeding during the mating season [81, 82]. Therefore, we suggest that the silky shark in Aceh may also exhibit special mating behavior or migratory patterns to reduce the chance of mating with close kin. However, further research on mating behavior is needed to provide additional evidence supporting this hypothesis. The potential effect by mixture of ancestry of the silky shark in Aceh population has also been evaluated by separating Aceh population into two sub-populations based on cluster I or cluster II they belong to (Fig 3) and reran the LD test, the results still showed a significant linkage for all comparisons of the cluster I sub-population (S6 Table). Therefore, the most likely explanation for the LD we observed is the small population size of Aceh. In small populations such as Aceh population in our case, the effect of genetic drift tends to be stronger than in larger populations, and the demographic fluctuation (i.e. population bottleneck) could also act as a potential mechanism to cause the findings we observed.

Ne of both the Pacific and the Indian Oceans exceeds 100, which is considered sufficient to avoid inbreeding in the population in the short-term period [83]. However, to maintain evolutionary potential over the long term, an Ne greater than 1000 is suggested [83]. Unfortunately, this Ne level was not observed in the silky sharks in either region during the current study. A previous study on the scalloped hammerhead *Sphyrna lewini* reported an effective population size [84] similar to that of the silky shark, emphasizing the importance of assessing its stock status. Consequently, our results further underscore the urgent need for the management and conservation of the silky shark. Additionally, the smaller Ne value observed in the Indian Ocean, in comparison to the Pacific Ocean, may be attributed to the high fishing pressure in the Indian Ocean. While there are no regulations on the silky shark catch in the Indian Ocean, catches and retention of the silky shark are prohibited in the Western and Central Pacific.

## Management implication and future work

In the present study, we have unveiled the contemporary genetic structure of the silky shark across the Indo-Pacific region, revealing the presence of multiple stocks in this area. This finding sharply contrasts with the "single stock" conclusion reached by Clarke et al. (2015), which relied solely on mtDNA markers [36]. Our results strongly suggest that the population inhabiting Aceh waters should be managed separately from the others. Furthermore, within the broadly distributed genotype group observed across the Indo-Pacific, a moderate genetic structure between populations in the Pacific and Indian ocean. Therefore, as more migratory data become available in the future, it may be more appropriate to consider them as two subpopulations.

Given the inconsistent management strategies in the Western and Central Pacific, and the Indian Ocean, it is crucial to understand the stock structure across these two regions. Unlike marine invertebrates and teleost fishes, sharks lack a planktonic larval stage, highlighting the importance of dispersal patterns and dispersal abilities of various life history stages in shaping their genetic connectivity [85]. To accurately identify shark stocks, it is essential to understand their movement patterns, distribution ranges, and the degree of spatial overlap with neighboring stocks. Additionally, shark behaviors such as philopatry, residency, and site fidelity can play a vital role in their stock structure [86–88]. Future studies focusing on tracking methods (i.e., satellite tagging) to assess their migratory and behavioral patterns can help delineate more precise boundaries for different stocks of the silky shark which could help design a better management plan to prevent local extinction.

## Supporting information

**S1 Table. Genotypes of 307 silky sharks among 5 sampling locations.**
(XLSX)

**S2 Table. Hardy-Weinberg equilibrium test based on 8 loci.** Estimation of exact P-Values by the Markov chain method. Bold indicated significant deviation from HW after Bonferroni correction.
(XLSX)

**S3 Table. Result of linkage disequilibrium between pair of loci based on 8 loci.** Bold values indicate significant departures ($P<0.05$) from Linkage disequilibrium, asterisk indicate significant departures after Bonferroni correction.
(XLSX)

**S4 Table. Pairwise $F_{ST}$ value among the 5 sampling sites (below diagonal) and corresponding p values (above diagonal) with 6 loci (exclude cafa16 and cafa44).**
(XLSX)

**S5 Table. Effective population size (Ne) for three groups: PO = Pacific Ocean; Aceh; IN = Indian Ocean (SLS, SLW, OB) with $P_{crit}$ = 0.02.** N = Harmonic Mean sample size with 6 loci (exclude cafa16 and cafa44).
(XLSX)

**S6 Table. LD results of using 6 loci and separating Aceh sample into Aceh_Cluster I and Aceh_Cluster II.** Bold values indicate significant departures ($P<0.05$) from Linkage disequilibrium, asterisk indicate significant departures after Bonferroni correction.
(XLSX)

**S1 Fig. Pie charts showing the proportion of ancestry assigned to individuals of each population (K = 2) by Bayesian clustering analysis based on 6 loci.**
(PNG)

## Acknowledgments

We would like to thank the Fisheries Agency of Taiwan for providing the tissue samples of silky sharks from confiscated silky shark catch. Ci Cheng, Shan-Hui Su, Yao-Yu Hsu, Chi-Hsuan Hsu, Xing-Han Wu, and crews from the National Kaohsiung University of Science and Technology for helping collect the tissue samples. Ms. Dilhara Wijetunge for assistance during the 2020 field trip in Sri Lanka.

## Author Contributions

**Conceptualization:** Wen-Pei Tsai, Shang Yin Vanson Liu.

**Data curation:** Chia-Yun Joanne Li, Wen-Pei Tsai, Shang Yin Vanson Liu.

**Formal analysis:** Chia-Yun Joanne Li.

**Funding acquisition:** Wen-Pei Tsai, Shang Yin Vanson Liu.

**Investigation:** Chia-Yun Joanne Li, Shang Yin Vanson Liu.

**Methodology:** Chia-Yun Joanne Li, Shang Yin Vanson Liu.

**Project administration:** Shang Yin Vanson Liu.

**Resources:** Chia-Yun Joanne Li, Wen-Pei Tsai, R. R. M. K. P. Ranatunga, Munandar Samidon.

**Supervision:** Shang Yin Vanson Liu.

**Validation:** Chia-Yun Joanne Li, Wen-Pei Tsai, Shang Yin Vanson Liu.

**Visualization:** Chia-Yun Joanne Li.

**Writing – original draft:** Chia-Yun Joanne Li.

**Writing – review & editing:** Chia-Yun Joanne Li, Shang Yin Vanson Liu.

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
