## [Decision Letter · Decision Letter 0]

6 Jun 2023

PONE-D-23-11283Genetic stock structure of the silky shark Carcharhinus falciformis in the Indo-Pacific OceanPLOS ONE

Dear Dr. Liu,

Thank you for submitting your manuscript to PLOS ONE. After careful consideration, we feel that it has merit but does not fully meet PLOS ONE’s publication criteria as it currently stands. Therefore, we invite you to submit a revised version of the manuscript that addresses the points raised during the review process.

Add details (table) on the sampling procedure and samples used.

The introduction seems a little too long and broad for the content of the paper, shorten please

We are concerned about the effective population size estimate, since sample from one populations was very small.

The authors are urged to significantly improve the language and style of the manuscript, which currently contains many linguistic, stylistic, and grammatical issues.

We look forward to receiving your revised manuscript.

Kind regards,

Arnar Palsson, Ph.D.

Academic Editor

PLOS ONE

“This study was funded by Ministry of Science and Technology, Taiwan (MOST) under No. MOST110-2313-B-992-001, No. MOST107-2911-I-110-301.”

3. We note that Figures 1,3 and 4 in your submission contain [map/satellite] images which may be copyrighted. All PLOS content is published under the Creative Commons Attribution License (CC BY 4.0), which means that the manuscript, images, and Supporting Information files will be freely available online, and any third party is permitted to access, download, copy, distribute, and use these materials in any way, even commercially, with proper attribution. For these reasons, we cannot publish previously copyrighted maps or satellite images created using proprietary data, such as Google software (Google Maps, Street View, and Earth). For more information, see our copyright guidelines: http://journals.plos.org/plosone/s/licenses-and-copyright.

a. You may seek permission from the original copyright holder of Figures 1,3 and 4 to publish the content specifically under the CC BY 4.0 license. 

Additional Editor Comments:

Add details (table) on the sampling procedure and samples used.

The introduction seems a little too long and broad for the content of the paper, shorten please

We are concerned about the effective population size estimate, since sample from one populations was very small.

The authors are urged to significantly improve the language and style of the manuscript, which currently contains many linguistic, stylistic, and grammatical issues.

Reviewers' comments:

Reviewer's Responses to Questions

**Comments to the Author**

1. Is the manuscript technically sound, and do the data support the conclusions?

Reviewer #1: Yes

Reviewer #2: Yes

2. Has the statistical analysis been performed appropriately and rigorously? 

Reviewer #1: Yes

Reviewer #2: Yes

3. Have the authors made all data underlying the findings in their manuscript fully available?

Reviewer #1: Yes

Reviewer #2: Yes

4. Is the manuscript presented in an intelligible fashion and written in standard English?

Reviewer #1: No

Reviewer #2: Yes

5. Review Comments to the Author

Reviewer #1: General comments:

The authors used 5 microsatellite loci to examine the population structure of silky sharks based on 307 processed filet tissue samples, collected at 5 different fish markets across the Indo-Pacific. I generally do not find any major flaws in the methodology but do find the observed, very strong genetic separation of samples from Aceh to all other locations very surprising. I would urge the authors to include more details on the sampling procedure and samples used in this study, to eliminate any potential doubts regarding the origin of their samples. In addition, the authors are urged to significantly improve the language and style of the manuscript, which currently contains many linguistic, stylistic, and grammatical issues.

Introduction:

The introduction seems a little too long and broad for the content of the paper. The authors should also spend some time on improving the flow of this section. At the moment there are several abrupt topical changes between individual paragraphs, which make it difficult for the reader to follow the overall story.

Line 43-45: Please use updated numbers, reference 3 is outdated.

Line 55-57: There are a lot more oceanic shark species than the ones mentioned, so better replace “i.e.” with “e.g.”

Line 110-111: Kraft et al. 2020 is missing in the reference list.

Materials and Methods:

Please include a description of how the Monmonier’s algorithm was used in your analysis as results for this method are presented in the results section but there is no mention in the methods section.

131-140: More details on sample collection need to be included, e.g. how were sampled sharks identified, what kind of tissue samples were collected, which part of the body was sampled, were samples collected from the whole animal, etc.

References:

Middle initials are missing for most authors in the reference list and must be included before publication, e.g. 41, 62, 64 and many more.

In-text citations are sometimes in name-year style, sometimes in number style.

Reviewer #2: In this study Chia-Yun Joanne Li and collaborators add a significant number of samples to analyze the population structure of the Silky shark in Indo-Pacific. By doing so they provide valuable and new information for the management of this species. The manuscript is technically sound and the data strongly support the conclusions. The methodologies and analyses were not cutting edge but all were conducted appropriately and conclusions are robust. Fairly minor issues that I think can be easily addressed by the authors with a minor revision. I congratulate all authors. Below I provide some detailed comments:

I have a small concern regarding the effective population size data, since one of the populations has a very small sample size that can be affecting your final results. The statistical analyses is rigorous and appropriate, again only one minor concern regarding the effective population size results. Also, the results from the the HW and linkage disequilibrium analyses of each locus were not presented.

Line 66-67

Review this sentence, you refer to the silky shark as plural and then as singular.

Line 72

Sometimes you refer to the silky shark as “the” silky shark and sometimes not, I suggest you are consistent throughout the paper.

Line 75

Sometimes you refer to the silky shark as “the” silky shark and sometimes not, I suggest you are consistent throughout the paper.

Line 103

The genetic population structure

Line 108

"if" the focal species instead of "while" the focal species

Line 109-111

Review this sentence

Line 110

This article is not on the reference list, do you refer to this article : Genomics versus mtDNA for resolving stock structure in the silky shark (Carcharhinus falciformis)

Line 146

Samples were amplified? I suggest you write down what were the reagents and their concentrations for the PCR reaction.

Line 149

PCR not PRC

Line 160

I suggest exact test in addition to Fst

Line 193

You do not present the results of HW and linkage disequilibrium for each locus

Line 201

Sampling locations plural

Line 232

Figure 4 I suggest you describe this barriers in the Figure heading to better understand the image.

Line 239

Be carefull with these samples sizes! I think they could be underestimating Ne in Aceh.

Line 296

Are there any other examples of this same pattern of genetic structure with other marine organisms?

Line 316-317

I see higher genetic diversity in Aceh, do you think this has to do with a higher number of samples from this location?

Also, the results of very low Ne from this location do not make sense with the genetic diversity found.

Line 339-342

You should check bias in estimates of Ne when samples sizes are low:

This is an article you should check out:

Estimating effective population size from linkage disequilibrium: severe bias in small samples Phillip R. England1,*, Jean-Marie Cornuet2 , Pierre Berthier3 , David A. Tallmon4 & Gordon Luikart5

Line 355

Sphyrna lewini not Sphyrna lewinin

6. PLOS authors have the option to publish the peer review history of their article (what does this mean?). If published, this will include your full peer review and any attached files.

Reviewer #1: No

Reviewer #2: No

---

## [Author Response · Author response to Decision Letter 0]

27 Jul 2023

Responses to Editor's comments:

Add details (table) on the sampling procedure and samples used.

The introduction seems a little too long and broad for the content of the paper, shorten please.

We are concerned about the effective population size estimate, since sample from one populations was very small.

The authors are urged to significantly improve the language and style of the manuscript, which currently contains many linguistic, stylistic, and grammatical issues.

R: We have rearranged and shorten the introduction and rests of the comments raised by the reviewers have been properly addressed, the revision is also polished by English editing company.

1. The adjustments have been made including the file's name and the manuscript.

2. The role of the funder in the study has been added to the cover letter” Additionally, the funders had no role in study design, data collection and analysis, decision to publish, or preparation of the manuscript.”

3. Figure 1, 3 and 4 were generated using ggOceanMap and ggplot2 packages and the citations of these two packages have been added. ggOceanMaps uses openly available geographic data and the source of the map (Natural Earth data) was also acknowledged.

Reviewer #1

The introduction seems a little too long and broad for the content of the paper. The authors should also spend some time on improving the flow of this section. At the moment there are several abrupt topical changes between individual paragraphs, which make it difficult for the reader to follow the overall story.

R: We have rearranged and rephrased the paragraphs to make it more concise.

Line 43-45: Please use updated numbers, reference 3 is outdated.

R: The numbers have been updated.

Line 55-57: There are a lot more oceanic shark species than the ones mentioned, so better replace “i.e.” with “e.g.”

R: The correction has been made.

Line 110-111: Kraft et al. 2020 is missing in the reference list.

R: The citation and reference have been added.

Please include a description of how the Monmonier’s algorithm was used in your analysis as results for this method are presented in the results section but there is no mention in the methods section.

R: We have added a description as follows “The barrier analysis was applied to detect the genetic boundary between the population based on Monmonier's (1973) algorithm (53) function in Adgenet v1.3.4 package (51) in R which detects boundaries among geographic populations with both geographic and genetic distances as valuated graph. This is achieved by finding the path exhibiting the largest distances between connected populations.” in the methods section. 

Line 131-140: More details on sample collection need to be included, e.g. how were sampled sharks identified, what kind of tissue samples were collected, which part of the body was sampled, were samples collected from the whole animal, etc.

R: More details on sample collections have been added as follows “A total of 307 tissue samples were collected from 5 locations, including the Southwestern Pacific Ocean (PO, N = 116), Indonesia Aceh fish market (Aceh, N = 41), Sri Lanka Negombo and Colombo fish market (SLW, N = 68), Mirissa and Kudawella fishing ports (SLS, N = 40), and Western Indian Ocean (OB, N = 42) (Figure 1) from September 2018 to January 2021. A small tissue of fin or muscle (1 cm3) was collected from each shark and preserved immediately in 95% EtOH.”

Middle initials are missing for most authors in the reference list and must be included before publication, e.g. 41, 62, 64 and many more.

& In-text citations are sometimes in name-year style, sometimes in number style.

R: The correction has been made.

Reviewer #2

I have a small concern regarding the effective population size data, since one of the populations has a very small sample size that can be affecting your final results. The statistical analyses is rigorous and appropriate, again only one minor concern regarding the effective population size results. 

&

Line 239

Be careful with these samples sizes! I think they could be underestimating Ne in Aceh.

R: I understand your concern and we had the same concern as well. In order to exclude this potential effect we ran a separate test by randomly reduce the sample size of Indian Ocean and Pacific Ocean to 41 and the results showed a similar figure as we got in the Table 4. Which indicate that the estimation of the Ne based on the genetic variations in Aceh population were referring by its potential low genetic diversity (isolated population).

Line 316-317 I see higher genetic diversity in Aceh, do you think this has to do with a higher number of samples from this location?

Also, the results of very low Ne from this location do not make sense with the genetic diversity found.

&

Line 339-342 You should check bias in estimates of Ne when samples sizes are low:

This is an article you should check out:

Estimating effective population size from linkage disequilibrium: severe bias in small samples Phillip R. England1,*, Jean-Marie Cornuet2 , Pierre Berthier3 , David A. Tallmon4 & Gordon Luikart5

R: Thank you for your valuable comment, the sample size of Aceh is the second lowest among 5 populations, therefore, it unlikely the higher Ho was due to higher number of samples in this population. However, we admit that during the pooling process the overall heterozygosity may change accordingly then could ultimately influence the Ne estimation. Therefore, we added a paragraph to address these potential influences and added the reference suggested by reviewer as follows “Although serval factors may affect the Ne that estimated in the present study, including the smaller sample size of Aceh (N=41) used may cause a potential underestimation of the Ne (78) and the potential bias caused by pooling samples from different geographic locations (Indian Ocean population). The Ne that we estimated in the present study may not represent the exact Ne; instead, it is a relative Ne value estimated based on the genetic variations in each population, which indicates that the silky shark inhabiting the Aceh waters could be an isolated and endemic population.” 

Line 66-67 Review this sentence, you refer to the silky shark as plural and then as singular.

R: The corrections have been made.

Line 72 Sometimes you refer to the silky shark as “the” silky shark and sometimes not, I suggest you are consistent throughout the paper.

R: The corrections have been made.

Line 75 Sometimes you refer to the silky shark as “the” silky shark and sometimes not, I suggest you are consistent throughout the paper.

R: The corrections have been made.

Line 103 The genetic population structure

R: The correction has been made.

Line 108 "if" the focal species instead of "while" the focal species

R: The correction has been made.

Line 109-111 Review this sentence

R: We rephrased it as “Nuclear markers are commonly used to discriminate the genetic structure of sharks (ie. Blackmouth catshark, 38), making them a valuable tool in stock identification studies.

Line 110 This article is not on the reference list, do you refer to this article : Genomics versus mtDNA for resolving stock structure in the silky shark (Carcharhinus falciformis)

R: The reference has been added.

Line 146 Samples were amplified? I suggest you write down what were the reagents and their concentrations for the PCR reaction.

R: We have added more info as follows “Each reaction contained 50ng DNA, 12.5μL TaqDNA polymerase 2X master Mix RED (0.4 mM each dNTPs, 15 mM MgCl, 0.2 unit of Ampliqon DNA polymerase) (Ampliqon Denmark) and each of primers (200nM). Genotyping was carried out by ABI sequencer.”.

Line 149 PCR not PRC

R: The corrections have been made.

Line 160 I suggest exact test in addition to Fst

R: We have revised the sentence as “Pairwise FST value with Fisher’s exact test was estimated using MSA 4.05”

Line 193 You do not present the results of HW and linkage disequilibrium for each locus

R: We have add the results of HWE and LD in S2 and S3 Table, respectively and added a paragraph to described the results as “The results of HWE test showed that most of the populations at locus Cafa16 and Cafa44 were deviated from HWE, and significant LD mainly occurred in Aceh across all loci (S2-S3 Table).”

Line 201 Sampling locations plural

R: The correction has been made.

Line 232 Figure 4 I suggest you describe this barriers in the Figure heading to better understand the image.

R: We added the following sentence in the Figure caption as “The thickness of blue lines represented the levels of the effect of barriers on gene flow.”

Line 296 Are there any other examples of this same pattern of genetic structure with other marine organisms?

R: So far, there were only few cases which focus on giant clam and seagrass showed population from Aceh were genetically different others by using microsatellite loci, but as for pelagic species our case seems to be the first one. Since the driving force for the genetic partitioning of those demersal species may be different from the pelagic species, therefore, we did not include these literatures in the discussion part.

Line 355 Sphyrna lewini not Sphyrna lewinin

R: The correction has been made.

---

## [Decision Letter · Decision Letter 1]

6 Sep 2023

PONE-D-23-11283R1Genetic stock structure of the silky shark Carcharhinus falciformis in the Indo-Pacific OceanPLOS ONE

Dear Dr. Liu,

Thank you for submitting your manuscript to PLOS ONE. After careful consideration, we feel that it has merit but does not fully meet PLOS ONE’s publication criteria as it currently stands. Therefore, we invite you to submit a revised version of the manuscript that addresses the points raised during the review process.

The manuscript has improved a great deal. There are two main concerns remaining however.

First, rev 2 points out LD in Aceh sampling site may affect the estimation of various population genetic statistics.

At first glance, this may be caused by mixture of ancestry of shark in that location (based on admixture results).

The effects of this could be tackled by subdivision of the Aceh sample and reestimation of statistics. Please discuss and respond.

Second, the language still requires more work. Rev 1 provides amble  editing suggestions, but many others can also be offered. Best if a third party reads this over and helps with grammar and clarity of text.

We look forward to receiving your revised manuscript.

Kind regards,

Arnar Palsson, Ph.D.

Academic Editor

PLOS ONE

Additional Editor Comments:

The manuscript has improved a great deal. There are two main concerns remaining however.

First, rev 2 points out LD in Aceh sampling site may affect the estimation of various population genetic statistics.

At first glance, this may be caused by mixture of ancestry of shark in that location (based on admixture results).

The effects of this could be tackled by subdivision of the Aceh sample and reestimation of statistics. Please discuss and respond.

Second, the language still requires more work. Rev 1 provides amble editing suggestions, but many others can also be offered. Best if a third party reads this over and helps with grammar and clarity of text.

Reviewers' comments:

Reviewer's Responses to Questions

**Comments to the Author**

1. If the authors have adequately addressed your comments raised in a previous round of review and you feel that this manuscript is now acceptable for publication, you may indicate that here to bypass the “Comments to the Author” section, enter your conflict of interest statement in the “Confidential to Editor” section, and submit your "Accept" recommendation.

Reviewer #1: (No Response)

Reviewer #2: All comments have been addressed

2. Is the manuscript technically sound, and do the data support the conclusions?

Reviewer #1: Yes

Reviewer #2: Partly

3. Has the statistical analysis been performed appropriately and rigorously? 

Reviewer #1: Yes

Reviewer #2: N/A

4. Have the authors made all data underlying the findings in their manuscript fully available?

Reviewer #1: No

Reviewer #2: Yes

5. Is the manuscript presented in an intelligible fashion and written in standard English?

Reviewer #1: No

Reviewer #2: Yes

6. Review Comments to the Author

Reviewer #1: Second round of review: Genetic stock structure of the silky shark Carcharhinus falciformis in the Indo-Pacific Ocean

While the authors have attended to most points raised by both reviewers, a few issues still remain:

Despite the authors employing an ‘English editing company’, concerns regarding language issues remain. These issues need to be fixed throughout the document, before the manuscript is ready for publication. I have just listed a few examples below to illustrate my concerns. Most mistakes are minor and can easily be fixed.

Line 19: replace “for” with “in”

Line 20-22: Here, we examined 307 individuals of silky sharks from 5 sampling locations across Indo-Pacific to reveal their genetic stock structure and effective population size by using eight microsatellite loci.

Should read:

Here, we used eight microsatellite loci to examine the genetic stock structure and effective population size of 307 silky sharks across 5 Indo-Pacific sampling locations.

Line 28: “We suggest”

Line 39-42: Check language and grammar - Elasmobranchs' k-selected life history characteristics (slow growth, late maturity, low fecundity, and long lifespan) have made them more vulnerable to overexploitation and have less potential for recovery than r-selected bony fish (1, 2).

Line 42-45: Check language and grammar - High global demands for shark products like fins, skins, gill plates, liver oil, and jaws has led to over 391 elasmobranch species being categorized as threatened by the International Union for Conservation of Nature (IUCN) (3).

Line 45-49: Check language and grammar - Numerous shark species have also been listed in the Convention on International Trade in Endangered Species of Wild Fauna and Flora (CITES) Appendix I and Appendix II, which restricted their international trade due to decreasing populations during the past few decades (4, 5). This situation urgently needs comprehensive management strategies for wild populations to reach the goal of future sustainable use.

Line 54: “patterns”

Line 55 Here and throughout the document: Please check the use of scientific and common names. As a general rule: At first mention, both the scientific and the common name, as well as the authority are given. After first mention you can choose whether to use the common or scientific name throughout the remainder of the document, but you must be consistent.

Line 61: sex-biased

Line 66-67: “In pelagic and coastal regions”

Line 73-76: Check language and grammar - “It is one of the most exploited shark species and the most common bycatch species of the tuna longline and purse seine fishery due to their overlapping representation with schools of tuna (16, 24, 25), their fins with high demand and abundant in the trade markets of Guangzhou, Hong Kong, and Taiwan (26-29).

Line 77-78: Check language and grammar - To date, the silky shark has been listed as a ‘Vulnerable’ species on IUCN Red List (30) and Appendix II species on the CITES to control the international trade of this species.

Line 96: add space

Line 107: Add “silky shark” before “population”

Line 115-116: Samples used in the present study are fish landing which were no longer alive when we obtained the tissue sample for population genetic analysis.

Line 258: “Karft” should read “Kraft”

Reviewer #2: I have one concern:

The results of HWE test showed that most of the populations at locus Cafa16 and Cafa44 were deviated from HWE, and significant LD mainly occurred in Aceh across all loci (S2-S3 Table). This is concerning. Do you think the LD is affecting your results? No loci with LD should be included in the analyses!! These results were not included in the initial submission that is why I had not addressed them earlier.

Statistical methods for analyzing HWE, FSTs, Ne ect.. assume independence between loci, so the presence of strong LD can violate this assumption and potentially lead to biased results!!

7. PLOS authors have the option to publish the peer review history of their article (what does this mean?). If published, this will include your full peer review and any attached files.

Reviewer #1: No

Reviewer #2: **Yes: **Mariana Elizondo-Sancho

---

## [Author Response · Author response to Decision Letter 1]

21 Sep 2023

Responses to Editor's comments:

Reviewer 2 points out LD in Aceh sampling site may affect the estimation of various population genetic statistics. At first glance, this may be caused by mixture of ancestry of shark in that location. The effects of this could be tackled by subdivision of the Aceh sample and reestimation of statistics. Please discuss and respond. 

R: We agree that the LD found in Aceh could possibly be due to the mixing of two genetic stocks, which aligns with what we mentioned in our discussion. However, if we separate Aceh into two sub-populations the sample size of each will be too small to perform further population genetic analyses. Therefore, in response to Reviewer 2’s comment and concerns specifically on the potential effect of loci Cafa16 and Cafa44 which deviated from HW. We excluded Cafa16 and Cafa44 and rerun Fst, Structure and Ne estimation, the results were similar to those based on 8 loci, except that the Ne value derived from 6 loci is smaller than that derived from 8loci for the PO population (Table S5). This suggests that adding these two loci didn't affect our observed patterns. Even if we separated these two subpopulations (cluster I and II), the results of LD test showed that cluster I from Aceh showed significant LD still for all the comparisons. Considering the small Ne we found in Aceh, this LD may be caused by genetic drift under a small population size as observed in the human population ( Laan and Pääbo, 1997). 

Reference: 

Laan, M., & Pääbo, S. (1997). Demographic history and linkage disequilibrium in human populations. Nature genetics, 17(4), 435-438.

The language still requires more work. Reviewer1 provides amble editing suggestions, but many others can also be offered. Best if a third party reads this over and helps with grammar and clarity of text. 

R: We appreciate the feedback and highlighting the concerns regarding the language issues in our manuscript. We have sent the revised manuscript for language editing again to improve the quality of this latest revision. 

Responses to Reviewer #1

Despite the authors employing an “English editing company”, concerns regarding language issues remain. These issue need to be fixed throughout the document, before the manuscript is ready for publication. I have just listed a few examples below to illustrate my concerns. Most mistakes are minor and can easily be fixed. 

R: We sincerely appreciate the reviewers’ comments regarding the language issue.

In this revision, we have corrected the issues mentioned by Reviewer 1 and then sent the revision to an English editing company for a comprehensive language editing service before submitting it back to PlosOne. We hope this version meets the publishing standard of PlosOne. 

Line 19: replace “for ” with “in”

R: The correction has been made.

Line 20-22: Here, we examined 307 individuals of silky sharks from 5 sampling locations across Indo-Pacific to reveal their genetic stock structure and effective population size by using eight microsatellite loci.

Should read:

Here, we used eight microsatellite loci to examine the genetic stock structure and effective population size of 307 silky sharks across 5 Indo-Pacific sampling locations.

R: The correction has been made.

Line 28: “We suggest ”

R: The correction has been made.

Line 39-42: Check language and grammar - Elasmobranchs' k-selected life history characteristics (slow growth, late maturity, low fecundity, and long lifespan) have made them more vulnerable to overexploitation and have less potential for recovery than r-selected bony fish (1, 2).

R: The sentence has been revised to : Due to their K-selected life history traits, including slow growth, late maturity, low fecundity and long lifespan, elasmobranchs are more susceptible to overfishing and have lower recovery potential compared to r-selected bony fish (1, 2)

Line 42-45: Check language and grammar - High global demands for shark products like fins, skins, gill plates, liver oil, and jaws has led to over 391 elasmobranch species being categorized as threatened by the International Union for Conservation of Nature (IUCN) (3).

R: The sentence has been revised to: High global demand for shark products, such as fins, skins, gill plates, liver oil, and jaws, has led to over 391 elasmobranch species being categorized as threatened by the International Union for Conservation of Nature (IUCN) (3)

Line 45-49: Check language and grammar - Numerous shark species have also been listed in the Convention on International Trade in Endangered Species of Wild Fauna and Flora (CITES) Appendix I and Appendix II, which restricted their international trade due to decreasing populations during the past few decades (4, 5). This situation urgently needs comprehensive management strategies for wild populations to reach the goal of future sustainable use.

R: These sentences have been revised to : Numerous shark species have also been listed in Appendix I and Appendix II of the Convention on International Trade in Endangered Species of Wild Fauna and Flora (CITES), restricting their international trade in response to declining populations over the past few decades(4, 5). This situation underscores the urgent need for comprehensive management strategies to ensure the future sustainable use of wild population.

Line 54: “patterns”

R: The correction has been made.

Line 55 Here and throughout the document: Please check the use of scientific and common names. As a general rule: At first mention, both the scientific and the common name, as well as the authority are given. After first mention you can choose whether to use the common or scientific name throughout the remainder of the document, but you must be consistent.

R: The correction has been made. Scientific and common names were provided for each species in the first mention. Subsequently, we used the common names throughout the manuscript. 

Line 61: sex-biased

R: the correction has been made.

Line 66-67: “In pelagic and coastal regions”

R: the correction has been made.

Line 73-76: Check language and grammar - “It is one of the most exploited shark species and the most common bycatch species of the tuna longline and purse seine fishery due to their overlapping representation with schools of tuna (16, 24, 25), their fins with high demand and abundant in the trade markets of Guangzhou, Hong Kong, and Taiwan (26-29).

R: These sentences have been revised to : It is among the most exploited shark species and is commonly caught as bycatch in tuna longline and purse seine fishery due to its overlapping presence with tuna school (16, 24, 25). Additionally, there is a high demand for its fins, which are abundantly traded in markets in Guangzhou, Hong Kong, and Taiwan (26-29).

Line 77-78: Check language and grammar - To date, the silky shark has been listed as a ‘Vulnerable’ species on IUCN Red List (30) and Appendix II species on the CITES to control the international trade of this species.

R: These sentences have been revised to :To date, the silky shark has been listed as a ‘Vulnerable’ species on IUCN Red List (30) and as an Appendix II species under CITES to regulate its international trade.

Line 96: add space

R: The correction has been made.

Line 107: Add “silky shark” before “population”

R: The correction has been made.

Line 115-116: Samples used in the present study are fish landing which were no longer alive when we obtained the tissue sample for population genetic analysis.

R: The sentence has been revised to : Samples used in this study were from fish landings; the specimens were no longer alive when we collected tissue samples for population genetic analysis.

Line 258: “Karft” should read “Kraft”

R: The correction has been made. 

Responses to Reviewer #2

I have one concern:

The results of HWE test showed that most of the populations at locus Cafa16 and Cafa44 were deviated from HWE, and significant LD mainly occurred in Aceh across all loci (S2-S3 Table). This is concerning. Do you think the LD is affecting your results? No loci with LD should be included in the analyses!! These results were not included in the initial submission that is why I had not addressed them earlier.

Statistical methods for analyzing HWE, FSTs, Ne ect.. assume independence between loci, so the presence of strong LD can violate this assumption and potentially lead to biased results!!

R: We appreciate the thoughtful feedback provided by the reviewer and have taken steps to address these concerns to ensure the robustness of our findings. To assess the potential impact of these two loci, we conducted an analysis using STRUCTURE, MSA, and Neestimator with the dataset excluded these two loci to examine the population structure patterns and the effective population size. 

Our analysis indicated that the exclusion of Cafa16 and Cafa44 did not substantially alter our results. 

1. STRUCTURE analysis: The bar plot generated from the STRUCTURE analysis exhibited a pattern consistent with the original analysis, suggesting that the population structure remain unchanged. 

2. Pairwise FST analysis: The results of the pairwise FST analysis were aligned with the patterns based on 8 loci. 

3. Effective population size: In the case of effective population size estimation, when employing the reduced data of 6 loci (excluding Cafa16 and Cafa44), we did observe a decrease in population size for the PO population (617.7 to 142.9). However, the other two populations exhibited a similar effective population sizes based on 8 loci. 

As we provided the results of those estimations in the supplementary material which could exclude the potential effect due to deviated from HW on Cafa16 and Cafa44, therefore, we would like to keep the original tables to provide a better resolution regarding the identification of genetic stock. 

In addition to the above replies, we have discussed the mechanisms may drive the results of LD in the section of effective population size as follow: 

This is further supported by the deviation from LD for all loci in the Aceh population, which could be influenced by several factors, including selection, genetic drift, inbreeding, demographic history and population subdivision (79). However, this situation is unlikely to be caused by natural selection and inbreeding in our case since the inbreeding coefficient of Aceh was the lowest (FIS= 0.02), suggesting that it could be a population with random mating. Theoretically, individuals in small and isolated populations tend to inbreed (80). Nevertheless, in some sharks (e.g., the sicklefin lemon shark Negaprion acutidens and the basking shark), specific mating behaviors have evolved to avoid inbreeding during the mating season (81, 82) Therefore, we suggest that the silky shark in Aceh may also exhibit special mating behavior or migratory patterns to reduce the chance of mating with close kin. However, further research on mating behavior is needed to provide additional evidence supporting this hypothesis. The potential effect by mixture of ancestry of the silky shark in Aceh population has also been evaluated by separating Aceh population into two sub-populations based on cluster I or cluster II they belong to (Fig 3) and reran the LD test, the results still showed a significant linkage for all comparisons of the cluster I sub-population (S6 Table). Therefore, the most likely explanation for the LD we observed is the small population size of Aceh. In small populations such as Aceh population in our case, the effect of genetic drift tends to be stronger than in larger populations, and the demographic fluctuation (i.e. population bottleneck) could also act as a potential mechanism to cause the findings we observed.

---

## [Editor Report · Decision Letter 2]

27 Sep 2023

Genetic stock structure of the silky shark Carcharhinus falciformis in the Indo-Pacific Ocean

PONE-D-23-11283R2

Dear Dr. Liu,

We’re pleased to inform you that your manuscript has been judged scientifically suitable for publication and will be formally accepted for publication once it meets all outstanding technical requirements.

Kind regards,

Arnar Palsson, Ph.D.

Academic Editor

PLOS ONE
---

## [Editor Report · Acceptance letter]

2 Oct 2023

PONE-D-23-11283R2 

Genetic stock structure of the silky shark *Carcharhinus falciformis* in the Indo-Pacific Ocean 

Dear Dr. Liu:

I'm pleased to inform you that your manuscript has been deemed suitable for publication in PLOS ONE. Congratulations! Your manuscript is now with our production department. 

Kind regards, 

on behalf of

Dr. Arnar Palsson 

Academic Editor

PLOS ONE